# Uncovering Novel Features of the Pc Locus in Horn Development from Gene-Edited Holstein Cattle by RNA-Sequencing Analysis

**DOI:** 10.3390/ijms232012060

**Published:** 2022-10-11

**Authors:** Huan Wang, Huabin Zhu, Zhihui Hu, Nuo Heng, Jianfei Gong, Yi Wang, Huiying Zou, Shanjiang Zhao

**Affiliations:** Key Laboratory of Animal Genetics, Breeding and Reproduction, Ministry of Agriculture and Rural Affairs, Institute of Animal Science, Chinese Academy of Agricultural Sciences, Beijing 100193, China

**Keywords:** Holstein bull, horn development, Pc, Tild-CRISPR/Cas9, RNA-seq

## Abstract

The Polled Celtic (Pc) mutation locus is a genetically simple single mutation that is the best choice for breeding polled cattle using gene editing. However, the mechanism of the Pc locus for regulating horn development is unclear, so we used gene editing, somatic cell nuclear transfer and embryo transfer to obtain polled Holstein fetal bovine (gestation time 90 days) with a homozygous Pc insertion (gene-edited Holstein fetal bovine, EH) and the wild-type 90 days Holstein fetal bovine (WH) as controls. The hematoxylin-eosin (HE) staining results showed that, compared to the WH, the EH horn buds had no white keratinized projections or vacuolated keratinocytes and no thick nerve bundles under the dermal tissue. Furthermore, DNA sequencing results showed that the Pc locus was homozygously inserted into the fetal bovine genome. A total of 791 differentially expressed genes were identified by transcriptome sequencing analysis. Enrichment analysis and protein interaction analysis results of differentially expressed genes showed that abundant gene changes after Pc insertion were associated with the adhesion molecule regulation, actin expression, cytoskeletal deformation and keratin expression and keratinization. It was also noted that the results contained several genes that had been reported to be associated with the development of horn traits, such as *RXFP2* and *TWIST1*. This study identified these changes for the first time and summarized them. The results suggested that the Pc mutant locus may inhibit neural crest cell EMT generation and keratin expression, leading to failures in neural crest cell migration and keratinization of the horn bud tissue, regulating the production of the polled phenotype.

## 1. Introduction

Growth of horns have been a common vertebrate development, especially for bovines, where they are diverse in form and have medical and application value. However, with the rapid development of farming, horns will cause injury to other cattle and farm staff on intensive farms. The sharpness of the horns can increase the risk of physical damage to vital areas such as the udder, which can reduce the lifetime economic benefit of the cow [1,2]. In today’s cluster farming environment, cattle no longer need horns to defend themselves against predators and the presence of horns has become a constraint to the high-quality development and safety of large-scale dairy farming [3]. Calves are usually de-horned at 7 to 10 days old for economic and welfare reasons, but the existing methods of dehorning are not only labor-intensive and cause varying degrees of stress and injury to the calf but are also contrary to animal welfare principles [4]. The breeding of polled (hornless) cattle has important applications for dairy farming and the development of the dairy industry.

Medugorac et al. (2012) first identified a single mutation associated with the polled trait in several European beef breeds, such as polled Angus and Galloway cattle [1]. This mutation site was located on chromosome 1 (BTA1) and is a duplication of a 212 bp (1,705,834–1,706,045 bp) sequence that replaced an insertion of the original 10 bp (1,706,051–1,706,060 bp) sequence and was named Polled Celtic (Pc).

Carlson et al. in 2016, successfully produced polled calves by editing the Pc locus with Transcription Activator-like (TAL) Effector Nucleases (TALEN). Subsequently, Young et al. selected one of polled calves and crossed it with a horned cow (pp) to obtain six heterozygous (Pcp) polled calves. The Pc allele segregated in the offspring of this bull, and inheritance of either allele produced polled calves [5,6]. Schuster et al. in 2020, successively edited the Pc locus in different breeds of cattle using Clustered Regularly Interspaced Short Palindromic Repeats (CRISPR) gene editing technology, to produce a polled cattle somatic cell line with a homozygous Pc insertion and successfully obtained polled calves [7]. Hennig et al. in 2022 used CRISPR/Cas9 to delete a 133 bp fragment containing 10 bp of the Pc locus that had been replaced. The experiment yielded genotypically 133 bp double allele knock-out fetuses, which were found to all exhibit horn bud development, indicating that removal of the 10 bp DNA sequence was not sufficient to result in a polled phenotype [8].

Until recently, the exact molecular mechanism through which the Pc mutation locus causes the polled trait in cattle was not known [9]. In a prior study, Tild-CRISPR/Cas9 was used to successfully obtain Pc homozygous inserted Holstein bull ear margin fibroblast cell lines based on Hennig SL’s study [10]. To investigate the molecular mechanisms by which the Pc mutation locus regulated the hornless trait and the molecular mechanisms of horn development in cattle, histomorphological and genomic identification, as well as transcriptome sequencing analyses, were performed on wild-type fetal bovine and prepared gene-edited fetal bovine horn bud tissues with Pc locus insertions.

## 2. Results

### 2.1. Somatic Cell Nuclear Transfer and Embryo Transfer Results

This laboratory successfully obtained a monoclonal cell line with a Pc locus homozygous insertion in the ear margins of Holstein bull by Tild-CRISPR/Cas9 technology. Both PCR detection and DNA sequencing identified that the Pc locus was homozygous and inserted into the genome of the cell line. The homozygous mutant cell was subsequently used as a donor cell and the donor was transplanted into a stage MII enucleated oocyte by somatic cell nuclear transfer technique to prepare reconstituted embryos and perform electrofusion to blastocysts in Figure 1A. A total of 120 reconstructed embryos were produced for this experiment, 91 of which were successfully electrofused. The cleavage rate on day 3 was 73% (66/91) and the blastocyst rate on day 7 was 37.5% (33/91), as shown in Figure 1B. Six and eight blastocysts were selected for transfer and quality testing, respectively, while the rest of the embryos were preserved using vitrification. The results of agarose gel electrophoresis and DNA sequencing showed that the Pc locus was homozygously inserted into the genomes of eight reconstituted embryos as shown in Figure 1C,D. At 35 days, the first pregnancy rate was 66.7% (4/6) and the second pregnancy rate was 33.3% (4/6), as shown in Figure 1E,F.

### 2.2. Gene Editing Fetal Bovine Identification Results

In this experiment, the horn buds of EH fetal bovines and WH Holstein fetal bovines obtained by slaughter were identified morphologically and histologically. A distinct white keratinized projection could be observed in the WH horn bud tissue at 90 days by morphological observation, while none was observed in the EH group, seen in Figure 2. Based on HE staining, the epidermis of the WH horn buds is thickened in Figure 3E and more than ten layers of keratinocytes are visible in Figure 3F,H, while thicker nerve bundles in Figure 3G and hair follicles in Figure 3D are visible below the horn bud tissue. In contrast, the EH horn buds were only found with two to three thin layers of keratinocytes in Figure 3I,J and no obvious nerve bundles or hair follicles were found. The EH horn bud histomorphology was consistent with the frontal skin in Figure 3C,K. The PCR identification and DNA sequencing of WH and EH fetal bovines showed that the Pc locus was homozygously inserted into the genome of the gene-edited fetal bovine, seen in Figure 4.

### 2.3. Differentially Expressed Proteins between WH and EH

To investigate the regulatory mechanisms of bovine horn development, cDNA libraries of WH and EH horn bud tissues were constructed, and raw data subsequently obtained by Illumina high-throughput sequencing, with raw reads ranging from 49,208,676 to 56,395,082 for each sample. After eliminating many low-quality bases, splice-containing, and N-containing reads from the raw data, 56,135,334 (99.54%) to 48,996,840 (99.57%) clean reads were obtained in the WH and EH horn bud tissues and the Q30 was greater than 92.5% in all groups.

Sequence comparison showed that all groups were localized to the bovine reference genome (*Bos taurus*, UMD 3.1) with more than 46,947,510 (91.5%) clean reads and more than 95.5% of the above reads were uniquely mapped to the bovine reference genome. Based on the expression information, the Pearson correlation coefficient between any two samples was higher than 0.987. The heat map results showed that the inter-group sample reproducibility was good in the WH group as shown in Figure 5A, and the biological replicates between samples met the analytical criteria.

The differential gene screening and analysis were performed by DE-Seq. The differential gene screening criteria were |log2FC| ≥ 1 and *p*-value ≤ 0.05. The results showed that a total of 791 differentially expressed genes (DEGs) were screened in WH and EH as shown in Figure 5B and the volcano plot results showed that there were 230 up-regulated genes and 561 down-regulated genes in the above DEGs as shown in Figure 5C (Appendix A).

### 2.4. Enrichment Analysis Results of Differentially Expressed Genes

The GO enrichment, KEGG pathway and Reactome enrichment analyses were performed to explore the molecular mechanisms of the Pc locus regulating the polled trait in Holstein, as shown in Appendix A. There were 1733 GO terms of biological process, 144 of cellular component and 193 of molecular function enriched significantly in Figure 6A. The top 20 biological processes partially associated with adhesion included biological adhesion (GO:0022610), cell adhesion (GO:0007155) and cell-cell adhesion (GO:0098609). The top 20 molecular functions included the entries related to calcium ions, the actin cytoskeleton and cell adhesion molecules, such as calcium ions, actin cytoskeleton and cell adhesion molecules, such as calcium ion binding (GO:0005509), actin binding (GO:0003779) and cell adhesion molecule binding (GO:0050839) and cellular component was also enriched for actin cytoskeleton-related terms.

Similarly, KEGG pathway analysis results showed that Focal adhesion (ko04510), ECM-receptor interaction (ko04512) and Calcium signaling pathway (ko04020) were also enriched significantly in Figure 6B. Apart from Laminin interactions (R-BTA-3000157), the top 20 Reactome terms that were significantly enriched included terms related to keratinization, including keratin filaments binding cell-cell adhesion complexes (R-BTA-6809393) and keratinization (R-BTA-6805567). Such enrichment results are consistent with phenotypic differences as shown in Figure 6C.

### 2.5. Protein Interaction Network Results

In this experiment, protein interaction networks were analyzed using the STRING protein interaction database (http://string-db.org, accessed on 30 August 2022) for differentially expressed genes in Figure 7A. Hub protein analysis was performed using Cytoscape (U.S. National Institute of General Medical Sciences (NIGMS), Bethesda, MD, USA) and the interactions network of the top 20 Hub proteins was mapped in Figure 7B. The results showed that most of these proteins were focused on cell adhesion molecule regulation, including ITGB2, ITGB3 and CDH1 and regulation of cellular actin, such as ACTA1, ACTN2, ACTC1, ACTA2 and MYL1.

### 2.6. qRT-PCR Validation

The relevant expression levels of nine randomly selected genes *RXFP2*, *TWST1*, *CDH1*, *ITGB3*, *ACTA1*, *ACTN2*, *ACTN3*, *ASXL3* and *SOX10* were verified by qRT-PCR to validate the expression levels of mRNAs and lncRNAs. The results showed that the qRT-PCR results were consistent with the sequencing results, as shown in Figure 8.

## 3. Discussion

The horns of Holstein cows are a natural part of livestock production, but their presence in the context of high-density dairy farming is extremely risky and can very easily lead to serious consequences, such as abortion in pregnant cows and udder damage in lactating cows, significantly reducing the lifetime benefits for the cow [11]. As the most widely farmed and intensive high-yielding dairy breed in the world, Holstein cows are widely distributed, with the USA and China having a total of 9 million and 4.5 million cows, respectively. Such a large base makes de-horning extremely demanding [3], so breeding polled cows is even more important for Holstein herds. However, traditional hybrid breeding has long lead times, which makes it hard to provide polled cattle for the industry in the short term, so molecular design breeding has the potential to significantly reduce the breeding time for polled cattle. There are multiple polled genetic variants in nature, one of which is a dominant genetic variant with a 202 bp duplication on chromosome 1 called the angular Pc locus [1].

The molecular mechanisms of Pc mutation types regulating the polled trait are currently unclear, so in this experiment, the Holstein ear margin fibroblast cell line with a homozygous Pc insertion prepared in advance was used as the donor cell. Embryos were obtained by somatic cell nuclear transfer and fetal bovines were slaughtered at 90 days of transfer with estrus as day zero. It should be noted that due to the low calving rate of gene-edited cattle, only one gene-edited fetal bovine was used for transcriptome sequencing in this study, based on the clarity of the homozygous insertion of the Pc locus and the polled phenotype of the fetal bovine. Comparative analysis of gene expression in horn bud tissue at 90 days for WH and EH Holstein fetal tissue was conducted by transcriptome sequencing. A total of 230 up-regulated and 561 down-regulated were obtained in EH compared to WH (Figure 5C), including several genes that have been reported to be associated with the polled trait, such as *RXFP2* and *TWIST1* [12,13].

The migration, proliferation and differentiation of neural crest cells were prerequisites for the formation of various tissues and organs [14]. Capitan et al. showed that horn development was the result of tissue differentiation in the ectoderm and mesoderm [15], where the neural crest cell migration and differentiation may be the basis for horn formation. Wang Yu et al. also found that the neural crest cell migration pathway played a key role in the development of horn origin, and inferred those ruminant horns had the same cellular origin as neural crest stem cells in the head [16].

Studies showed that neural crest cells needed to undergo epithelial cell-cell contacts, lysis and other EMT events to initiate migration [17]. In this process, a series of adhesion molecules such as cadherin, laminin, integrins and the immunoglobulin superfamily, participated in the change [18]. The conversion of E-cadherin to N-cadherin is considered to be a marker of the initiation of cell migration [19]. This change resulted in a loss of cell polarity and reduced contact with surrounding cells and matrix, along with enhanced cell migration and motility, which provided the preconditions for neural crest cell migration [20]. Integrins, another class of Ca^2+^ dependent cell adhesion molecules, played an important regulatory role in the EMT response by providing connections and mediating interactions between cells, between cells and the extracellular matrix and between the extracellular matrix and each other [21]. Laminin is a family of large molecular weight glycoproteins that accumulated in the ECM, which stimulated cell adhesion, cell motility and regulated cell growth, migration, and differentiation during cell development [22]. The GO, KEGG and Reactome enrichment analyses of differentially expressed genes suggested that multiple cell adhesion and adhesion molecule related pathways were enriched and involved dozens of differentially expressed genes, including L1 cell adhesion molecule (*L1CAM*), laminin subunit gamma 2 (*LAMC2*), and laminin subunit alpha 5 (*LAMA5*) (Figure 6, Appendix A). The protein encoded by *L1CAM* is an immunoglobulin superfamily cell adhesion molecule whose increased expression led to reduced E-cadherin expression, which in turn disrupted E-cadherin-mediated adhesion junctions and increased cell motility [23]. The genes *LAMC2* and *LAMA5* were also involved in the regulation of EMT. Studies in a variety of cells showed that the silencing of *LAMC2* expression inhibited the initiation of EMT, which in turn led to blocked cell migration [24,25].

The results of this experiment revealed that the expression of genes expressing immunoglobulin adhesion molecules, such as *L1CAM*, *LAMC2*, and *LAMA5*, was significantly lower in EH horn buds compared to WH (Appendix A), suggesting that EMT was inhibited in EH horn buds, which may lead to failure of neural crest cell migration initiation. Similarly, the top 20 Hub proteins from the protein interaction analysis also screened for multiple proteins associated with cell adhesion, such as cadherin 1 (CDH1), integrin subunit beta 2 (ITGB2) and integrin subunit beta 3 (ITGB3) were involved in the biological process of the regulation of cell adhesion molecule in these results [26,27,28]. It suggested that the Pc locus may inhibit EMT in neural crest cells by regulating the expression of cell adhesion molecules, particularly E-cadherin.

During EMT, when cell adhesion molecules are altered, the actin structure begins to reorganize and the cytoskeleton and shape change, so that the cells gain motility and invasion [29]. Actin is the key structural protein that forms the cytoskeleton of cells and plays a major role in functions including division, migration, and vesicular transport, participating in the coalescence and contraction of the cellular microfilament skeleton, and providing the impetus for cell migration [30]. This study found both multiple terms involved in cytoskeleton- and actin-related molecular functional enrichment analysis and multiple differentially expressed proteins associated with cytoskeleton and actin in the top 20 Hub protein interactions, including actin alpha 1 (*ACTA1*), actinin alpha 2 (*ACTN2*), actin alpha cardiac muscle 1 (*ACTC1*), and actin alpha 2, smooth muscle (*ACTA2*). The *ACTA1*, *ACTC1*, and *ACTA2* genes encode one of six different actin proteins and Hu et al. demonstrated that overexpression of *ACTA1* inhibited cell proliferation and migration [31]. Knockdown of *ACTC1* significantly inhibited cell migration [32]. Zhang et al. (2020) found that down-regulation of *ACTA2* inhibited migration of neural stem cells by increasing Rho A expression and decreasing Rac1 expression to hinder actin filament polymerization [16]. In this study, the expression of many genes regulated with actin, including *ACTA1*, *ACTN2*, *ACTC1*, and *ACTA2*, was down-regulated in EH horn buds (Figure 7B, Appendix A). The difference in gene expression may result in the inability to produce sufficient alpha-actin to participate in the formation and deformation of the cytoskeleton of microfilaments, inhibiting neural crest cell migration and leading to the appearance of a keratinous trait.

This study also screened for key genes involved in regulating the interaction of cell adhesion molecules with cytoskeletal assemblies, such as the pleckstrin homology domain containing A7 (PLEKHA7). This is a regulator of the interaction between E-cadherin and the cytoskeleton, regulating cytoskeletal remodeling and cell polarization by participating in the linkage between the two and promoting cell motility [33]. The expression of PLEKHA7 was significantly down-regulated in EH horn bud tissue, suggesting that migration of neural crest cells in EH horn bud tissue may be inhibited compared to WH.

The above results suggested that the Pc locus may inhibit the development of EMT in neural crest cells by regulating the expression of cell adhesion molecules, preventing neural crest cells from initiating migration. At the same time, the insertion of the Pc locus may also inhibit the expression of actin and cytoskeletal deformation in neural crest cells, limiting the migration of neural crest cells. The neural crest cells are unable to migrate to the site of the horn bud to proliferate and differentiate into neuronal and mesenchymal cells required for horn development and due to the lack of a material basis for horn development, mutations at the Pc locus led to the creation of the polled trait.

Studies of horn development found that keratin may play a key role in it. A brief white keratinized projection appears on the horn bud tissue during development in horned fetal calves and disappears later in development, whereas no keratinized projection appears on the horn bud tissue throughout the development of hornless fetal calves [11]. It suggested that the keratinization process at the horn bud may be necessary for the development of the calf’s horn after birth. This study did not find white protrusions or vacuolated keratinocytes in the EH horn bud tissue (Figure 2), unlike WH horn bud tissue and this was consistent with the reported results [34]. Two terms related to keratinization were found among the top 20 results of Reactome enrichment analysis of the two groups of differentially expressed genes (Figure 6C), including keratin filaments bind cell-cell adhesion complexes (R-BTA-6809393) and keratinization (R-BTA-6805567). These terms contained many differentially expressed genes encoding keratin, such as keratin 4 (KRT4), keratin 15 (KRT15), keratin 17 (KRT17), keratin 19 (KRT19), and keratin 80 (KRT80), all of which showed high expression in WH horn bud tissues and low expression in EH horn bud tissues [35]. This study also enriched the genes encoding structural proteins that link the desmosomes of keratinocytes and epidermal keratinized envelope of keratinocytes, such as periplakin (PPL), which plays important role in cellular adhesion complexes supporting and cytoskeletal integrity supplying [36], envoplakin (EVPL), which forms a component of desmosomes and the epidermal cornified envelope [37], etc. In EH horn bud tissue, low expression of all these genes directly affected the keratinization process of the cells. It suggested that the homozygous insertion of the Pc locus may directly lead to a reduction in keratin content and failure of keratinization in fetal bovine horn bud tissue, failing to provide the basis for later horn development and leading to the emergence of the polled trait.

Based on the above experimental results, it seems that the Pc locus completed the regulation of key events in the formation of the polled trait early in fetal bovine development. The possible regulatory mechanisms include two points. First, the Pc locus may initiate the EMT event by regulating the expression of cell adhesion molecules, actin, and cytoskeletal deformation in neural crest cell migration, resulting in a decrease in the ability of neural crest cells to migrate. As a result, large numbers of neural crest cells are unable to migrate successfully to their designated sites at the horn buds for value-added differentiation. Secondly, the Pc locus may inhibit the expression of keratin at the horn bud and the process of keratinization by keratinocytes, failing to provide the material basis for later horn development. Based on the above process, we have drawn a hypothetical model of Pc regulation, see Figure 9. Numerous studies showed that the development of the bovine horn was based on ossification centers in the dermis and subcutaneous tissues, which gradually form after ossification. It is hypothesized that the ossification centers of the developing horn in postnatal calves are composed of nerve bundles formed by migrating and differentiated nerve cells from neural crest cells.

## 4. Materials and Methods

### 4.1. Sample Collection and Preparation

The donor cells for somatic cell nuclear transplantation were mutant cell lines with homozygous insertion of the Pc locus mediated by Tild-CRISPR/Cas9 technology in this laboratory. In previous experiments, we obtained the above cell lines carrying the Pc variant by the following steps: (1) First, we obtained the ear marginal tissue of a Holstein bull and established cell lines. (2) Second, the sgRNA was designed for the Pc locus controlling the polled in cattle, and its mutation efficiency was detected by T7El digestion and sequencing. (3) Third, the sgRNA was designed for the Pc locus controlling the polled form in cattle, and its mutation efficiency was detected by T7El digestion and sequencing. The PUC57 vector was used as the skeleton to produce homologous recombinant fragment through connecting the Pc fragment and homology length of 1600 bp (800 bp homology on each arm), then the homologous recombinant fragment was obtained by double enzyme digestion and sgRNA were co-transfected into the fibroblasts of the auricular margin in Holstein bulls. Finally, the Holstein ear margin fibroblast cell line with the homozygous Pc insertion was successfully obtained [10]. Ovaries and wild-type fetal bovines from Holstein cows were obtained from the same slaughterhouse (Dachang Slaughterhouse, Langfang, Hebei Province, China) and wild-type fetal bovines were selected according to the following principles: that they were Holstein and each fetal bovine had similar age and body condition. After selection, three 90 days old Holstein male fetal bovines were chosen as test animals. All recipient females for embryo transfer were young cows from Liangshan Kelong Herding, Shandong, China.

The results of a study by Aldersey et al. in 2020 showed that fetal bovines exhibit significant differences in horn traits at 90 days of gestation [34]. In horned cows, white projections as thickened vacuolated cells appear on the horn buds and form thick subcutaneous nerve bundles, so a slaughter date at 90 days of gestation was selected to obtain fetal bovines. Three wild-type 90 days male Holstein fetal bovines (WH) and one 90 days gene-edited polled fetal bovine (EH) were surgically processed to isolate horn bud tissue, which was then partially soaked in 10% paraformaldehyde and partially fast-frozen in liquid nitrogen at −196 °C for transport to the laboratory, for storage at −80 °C until RNA was isolated. All animal procedures used in this study were approved by the Animal Care and Use Committee of the Institute of Animal Sciences of Chinese Academy of Agricultural Sciences (IAS2021-19).

### 4.2. Somatic Cell Nuclear Transplantation

Homozygous mutant cell lines were cultured for two to three generations in 15% Dulbecco’s modified eagle medium (DMEM, 15% fetal bovine serum, 84% DMEM, 1% PS, Gibco, Waltham, MA, USA).

Slaughterhouse ovaries were washed repeatedly with saline supplemented with penicillin and streptomycin at 27 °C. In sterile conditions, cumulus-oocyte complexes (COCs) were aspirated from 3 to 6 mm follicles using a vacuum peristaltic pump with an 18-gauge needle. Selection of COCs with homogeneous cytoplasm and dense oocytes was made using a home-made oral pipette under the microscope with a heated stage. After washing the selected COCs twice with HEPES-buffered TCM199 (Sigma, St. Louis, MO, USA) medium and three times with culture media, the COCs were transferred into the culture media and incubated for 22 h in an incubator at 38.5 °C and 5% CO_2_ saturation humidity.

The COCs cultured to maturity were digested with 1% hyaluronidase and oocytes with first polar bodies, intact morphology and homogeneous cytoplasm were selected for transfer into the manipulation solution. The microscopic manipulation aspirated the first polar body and a little of the surrounding cytoplasm. The donor cells were injected into the enucleated oocytes at the denuded pinhole and subsequently transferred into the electrical fusion medium for 1 min. The oocytes were arranged in batches in the cell electrofusion bath in a uniform manner while ensuring that the electric field was perpendicular to the contact surface of both cells. The parameters of the CF150B Cell fusion instrument (BLS, Budapest, Hungary) were adjusted to a field strength of 2.4 kV·cm^−1^ and pulses of 10 µs and two pulses. The reconstituted embryos were transferred into the embryo culture medium and incubated in the incubator for 30 min. The reconstituted embryos were transferred to the A23187 solution for 5 min, then transferred to the 6-DMAP solution and incubated in the incubator for 6 h. The reconstituted embryos were then transferred to the embryo culture solution and the blastocyst formation was counted after 7 days in the incubator. The reconstructed embryos were finally transferred into the embryo culture medium, and the blastocyst formation was counted after 7 days of culture in the incubator.

### 4.3. Embryo Transfer

Young heifers of 14 to 15 months of age were selected for uniform body structure, good body condition, good genetic performance, and the presence of the corpus luteum on the ovaries on rectal examination. Recipient heifers were treated with 6 mL intramuscular cloprostenol for estrus synchronization, followed by observation and recording of estrus over 2 to 3 days.

At day 7 of embryo culture, embryos were graded according to embryo quality and quality blastocysts were stored in separate 0.25 mL plastic tubes in units of one embryo for transport. The remaining blastocysts were preserved by cryopreservation by vitrification. Referring to the report of Roper et al. (2018), the best recipient cattle were selected for embryo transfer in this experiment [38]. Ultrasound monitoring of pregnancy of recipient cows was conducted at around 30 and 90 days after blastocyst transfer into the recipient cow.

### 4.4. Embryo Genotyping Analysis

The experiment randomly selected eight embryos labelled a to h to extract the whole genome of single cell clones using the cell tissue blood whole genome DNA extraction kit (Tiangen, Beijing, China). The whole genome was used as the template and btHP-1963-F/btHP-1963-R shown in Appendix A was used as primer for polymerase chain reaction (PCR) amplification. The 25 μL PCR reaction system contained 12.5 μL of 2 × Phanta Max Master Mix (Vazyme, Nanjing, China), 1 μL of target fragment, 1 μL each of 100 mmol/L upstream and downstream primers and 9.5 μL of distilled water (ddH_2_O). The PCR reaction conditions included pre-denaturation at 93 °C for 30 s, denaturation at 93 °C for 20 s, annealing at 58 °C for 30 s and extension at 72 °C for 70 s, for 33 cycles; followed by an extension at 72 °C for 2 min and storage at 4 °C. The PCR products were identified by agarose gel electrophoresis (AGE) at a concentration of 1.2% and the amplified products were sequenced by Sanger at Huada Genetics (Huada, Shanghai, China).

### 4.5. Phenotypic Analysis of the Fetus

Horn bud tissue was obtained from the WH and EH Holstein male fetal bovines by vertical cutting. Tissue samples were fixed using 10% formalin and dehydrated in a graded ethanol series. They were washed in xylene and embedded in paraffin and 4 μm microtome sections were stained with hematoxylin and eosin (HE). Digital image acquisition was with a Prog Res C5 camera (Analytik Jena AG, Jena, Germany).

### 4.6. Genomic Analysis of the Fetus

The whole genome of the skin tissue was extracted separately using the cell tissue blood whole genome DNA extraction kit (Tiangen, Beijing, China) and was used as a template and NEW-F/NEW-R as primers as shown in Appendix A for PCR amplification. The PCR reaction system and reaction conditions were the same with embryo genome analysis. The PCR products were identified by AGE at a concentration of 1.2% and the amplified products were sequenced by Sanger at Huada Genetics (Huada, Shanghai, China).

### 4.7. RNA Extraction and Qualification

The TRIzol^®^ Reagent kit (Thermo Fisher, Waltham, MA, USA) was used to extract tissue RNA. Quality-qualified Total RNA was selected as the starting sample for mRNA sequencing. The quality required was RIN ≥ 7, the ratio of 28S to 18S RNA was greater than or equal to 1.5:1 by Agilent 2100 BioAnalyzer (Shanghai, China) and the starting volume required was in the range of 2 to 3 μg. The Qubit RNA assay kit (Thermo Fisher, Waltham, MA, USA) provided accurate quantification of starting Total RNA. The Ribo-Zero™ Magnetic Kit (Epicentre, Beijing, China) was used for the isolation and removal of rRNA from Total RNA. Quality control of rRNA removed from total RNA was performed using agarose gel electrophoresis, Nanodrop microspectrophotometer (Thermo Fisher, Waltham, MA, USA), and Agilent 2100 (Agilent, Santa Clara, CA, USA).

### 4.8. Quality Control

The raw reads from the downstream machine were quality controlled using fastp [39] to filter the low quality data to obtain clean reads, including reads with more than 5% N content, reads with more than 30% of bases in the read with a quality value of no more than 20 and reads with splice sequences removed to obtain clean reads that could be used for subsequent comparisons. The statistics and the clean reads could be used for subsequent comparisons, statistics, and analysis.

### 4.9. Differential Expression Analysis

The clean reads were compared to the reference genomic sequence using HISAT2 2.1.0 software (Institut Pasteur, Paris, France) [40] with an overall comparison rate of ≥70% for clean reads and ≤10% for multiple comparisons. Differential analysis was performed on all subgroups using DE-Seq 1.28.0 [41] software (Illumina, San Diego, CA, USA) with differential gene screening criteria of |log2FC| ≥ 1 and *p*-value ≤ 0.05. Hierarchical clustering analysis was also done on the screened differentially expressed genes using R 4.1.2 software (R Foundation for Statistical Computing, Vienna, Austria) [42] and genes with the same or similar expression behavior would be clustered together.

### 4.10. Bioinformatics Analysis and Statistical Analysis

In this experiment, Gene Ontology (GO) analysis of the differentially expressed genes were performed using R 4.1.2 software (R Foundation for Statistical Computing, Vienna, Austria) [43]. The GO terms and pathways with a *p*-value < 0.05 were significantly enriched. Kyoto Encyclopedia of Genes and Genomes (KEGG) enrichment analysis was performed using Kobas 3.0 (Peking University and Institute of Computing Technology, Beijing, China) software to obtain the enrichment of differentially expressed genes in the pathway. Differentially expressed genes were mapped to each term of the Reactome database (https://reactome.org/, accessed on 20 August 2022) and the number of differential genes per term was calculated to obtain a list of differential genes with Reactome function and a count of the number of differential genes. The experiment applied a hypergeometric test to identify Reactome terms that were significantly enriched in differential genes compared to the whole background. A STRING diagram (http://string-db.org, accessed on 30 August 2022) was constructed for interaction networks of differentially expressed proteins. Visual protein interaction network maps and Hub protein interaction analysis were produced using Cytoscape software (U.S. National Institute of General Medical Sciences (NIGMS), Bethesda, MD, USA).

### 4.11. qRT-PCR

Total RNA extracted from tissues was synthesized using the Revertra ace qPCR kit (Toyobo, FSQ-101, Osaka, Japan) according to the manufacturing instructions. The complementary DNA (cDNA) was synthesized using a QuantStudio5 real-time fluorescent quantitative PCR system (Applied Biosystems, Waltham, MA, USA) in a total volume of 15 μL. The reaction system consisted of 10 μL 2 × SYBR green PCR buffer (ABI, 4368708, Raleigh, NC, USA), 1 μL of 10 μM PCR forward primer, 1 μL of 10 μM PCR reverse primer, 1 μL of template, and 6 μL of de-RNAase water. The reaction procedure was one cycle, denaturation at 50 °C for 2 min, denaturation at 95 °C for 2 min, followed by 40 cycles of amplification, denaturation at 95 °C for 15 s, and annealing extension at 60 °C for 1 min, followed by melt curve analysis in triplicate. The expression data were normalized to glyceraldehyde-3-phosphate dehydrogenase (*GAPDH*) expression using the 2^−ΔΔCT^ method. The sequences of the primers used are shown in Appendix A.

### 4.12. Statistical Analysis

We used SAS software version 9.2 (SAS, Cary, NC, USA) for statistical analysis. Groups were compared by one-way analysis of variance (ANOVA) followed by the Duncan post-hoc test for multiple comparisons. *p* < 0.05 was considered statistically significant. Results are shown as means ± standard deviation (SD). All experiments were repeated at least in three independent experiments.

## 5. Conclusions

The present research will facilitate further studies on horn development and provide further insight into the molecular mechanisms of Pc locus regulation of the polled trait. The present research will provide ideas and a theoretical basis for exploring the mechanism of horn development and Pc locus regulation in Holstein cattle and provide methods for breeding polled Holstein bulls to improve the safety and economic efficiency of cattle farm breeding.

## Figures and Tables

**Figure 1 ijms-23-12060-f001:**
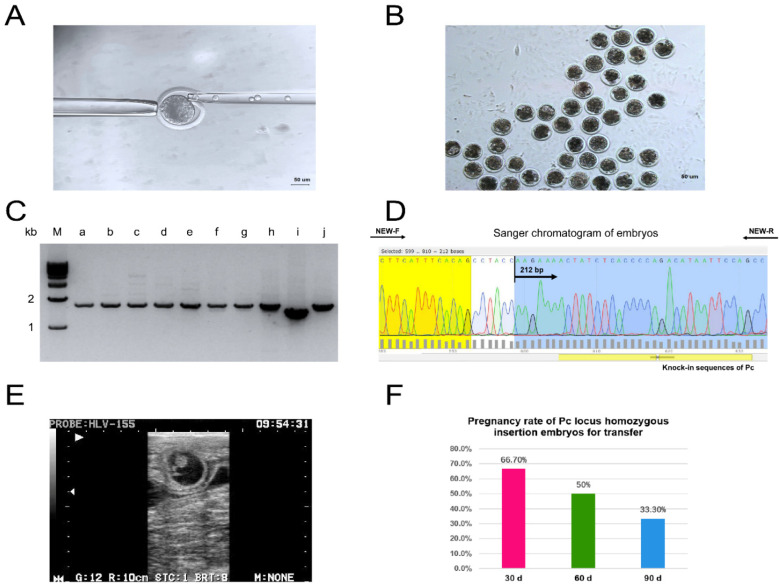
Preparation and identification of polled bovine embryos. (**A**) Transplantation of donor cells into enucleated oocytes. (**B**) Somatic cell nuclear transfer embryos in culture for 7 days. (**C**) PCR identification of blastocyst agarose gel electrophoresis results. (a–h) are eight somatic cell clones obtained from Pc site-pure insertions of hornless phenotypic blastocysts, (i) is the whole genome of wild-type horned Holstein, (j) is a cell line obtained from gene-edited Pc site homozygous insertions. (**D**) Sanger sequencing chromatograms of the Pc locus in somatic cell nuclear transfer embryos. (**E**) Ultrasound test results 30 days after embryo transfer. (**F**) Pregnancy rate of Pc locus homozygous insertion embryos for transfer.

**Figure 2 ijms-23-12060-f002:**
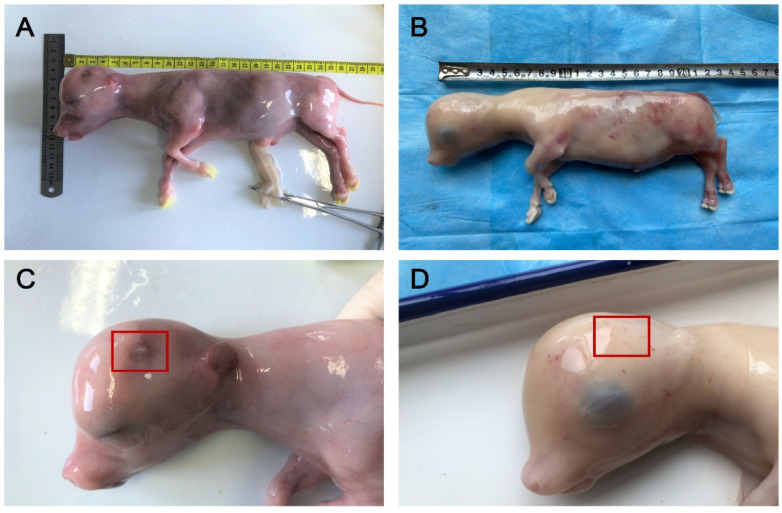
Phenotypic analyses of fetuses. (**A**,**C**) are horned Holstein 3-month-old fetuses, (**B**,**D**) are polled Holstein 3-month-old fetuses obtained by gene editing a pure insertion at the Pc locus from the enclosed horn bud sites in (**C**,**D**), it can be seen that there is distinct horn bud tissue at the horned horn buds.

**Figure 3 ijms-23-12060-f003:**
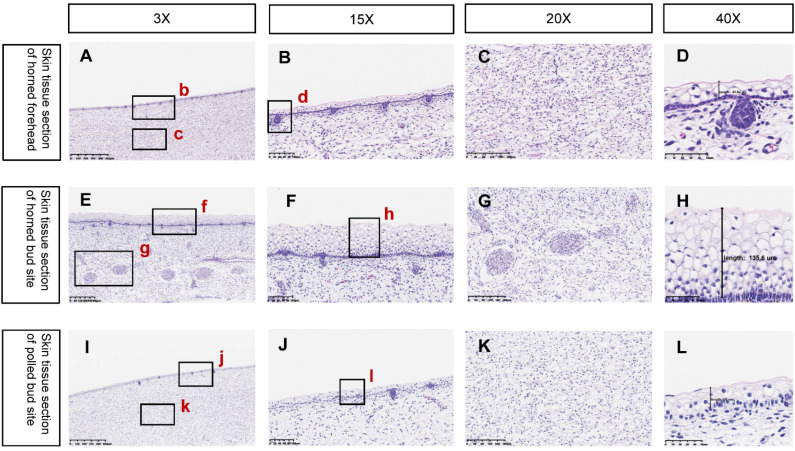
Histological identification of the fetus. (**A**–**D**) is wild-type frontal skin tissue with normal thickness epidermis, only 2–3 layers of vacuolated keratin-forming cells, epidermis with follicular growth, no thickened nerve bundles found under the skin. (**E**–**H**) is wild-type skin tissue at the horn bud with more than 10 layers of vacuolated keratin-forming cells and no follicular growth in the epidermis. There are thickened nerve bundles under the skin. (**I**–**L**) is skin tissue at the horn buds of polled cattle, with a normal thickness of epidermis visible, only 2–3 layers of vacuolated keratin-forming cells, and no thickened nerve bundles found under the horn buds. (**A**,**I**) magnification 3×, (**E**) magnification 5×, (**B**,**C**,**F**,**G**,**J**,**K**) magnification 10×, (**D**,**H**,**L**) magnification 30×.

**Figure 4 ijms-23-12060-f004:**
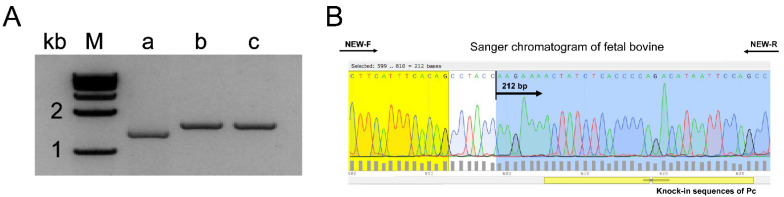
Fetal cow genome test results. (**A**) Results of agarose gel electrophoresis of PCR amplification products from gene edited hornless bovine fetuses. (a) Whole genome of Holstein 3-month-old horned fetus. (b) Cell line obtained by pure insertion of gene editing Pc locus. (c) Whole genome of Holstein 3-month-old polled fetus obtained by pure insertion of gene editing Pc locus. (**B**) Sanger chromatogram of fetal bovine with the Pc locus homozygous insertion.

**Figure 5 ijms-23-12060-f005:**
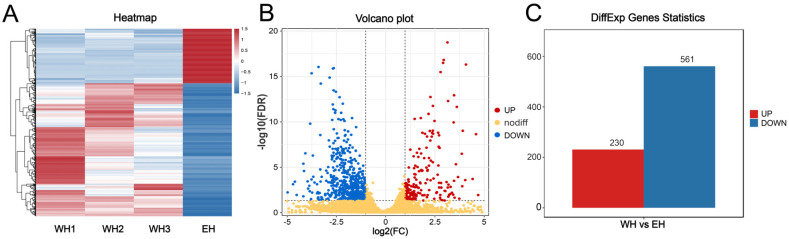
Sample gene expression. (**A**) Sample Correlation Heat Map. (**B**) Volcano plot of WH and EH differentially expressed genes. (**C**) Histogram of WH and EH differentially expressed genes.

**Figure 6 ijms-23-12060-f006:**
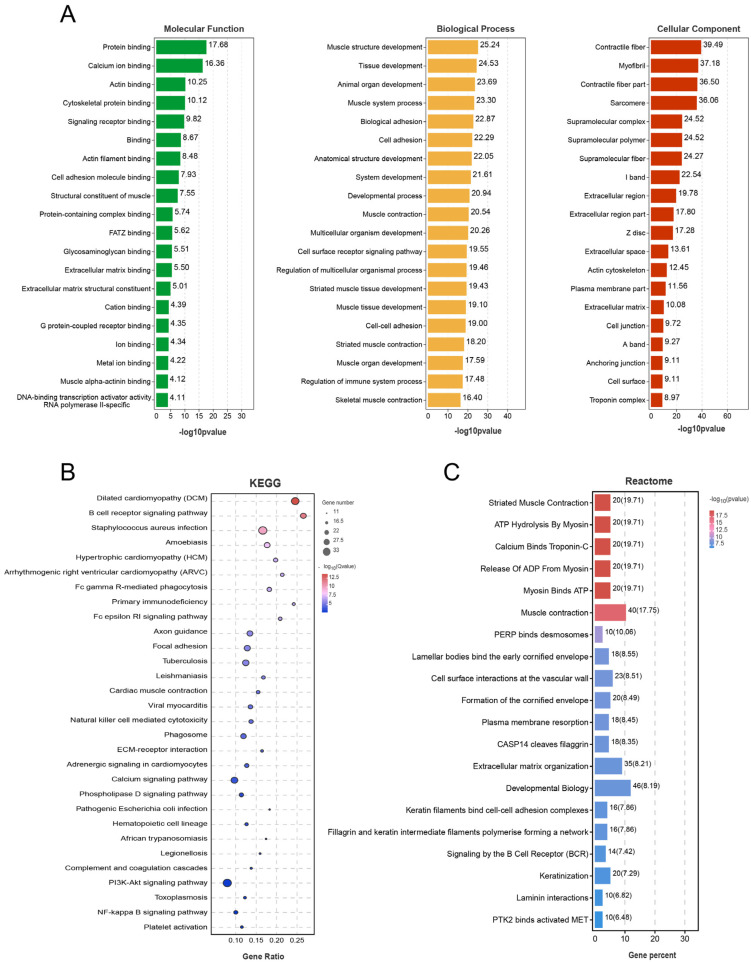
Results of differentially expressed genes enrichment. (**A**) GO enrichment analysis bar chart. (**B**) KEGG enrichment factor map. (**C**) Reactome enrichment analysis bar chart. Significant enrichment analysis was performed based on all significantly DE genes.

**Figure 7 ijms-23-12060-f007:**
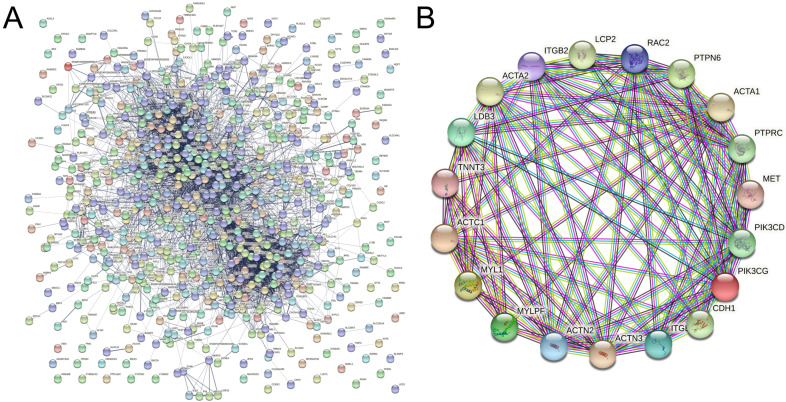
Results of differentially expressed protein interactions analysis. (**A**) Diagram of differentially expressed protein interaction network. (**B**) Hub protein interaction network diagram.

**Figure 8 ijms-23-12060-f008:**
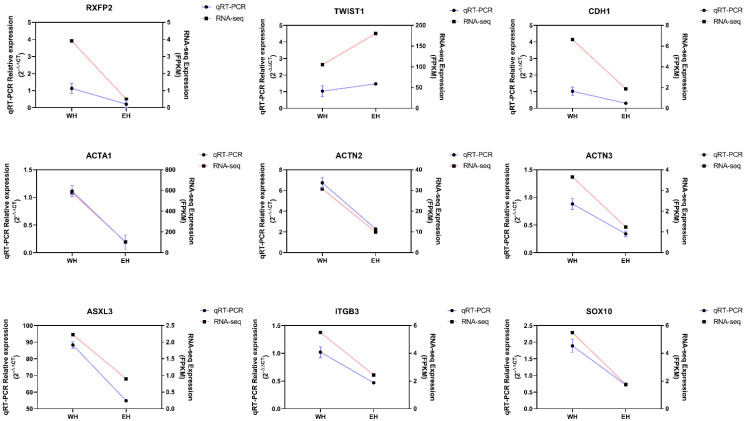
RT-qPCR validation for genes expression. The red line in the graph shows the results of RT-qPCR, the blue line is the result of RNA-seq.

**Figure 9 ijms-23-12060-f009:**
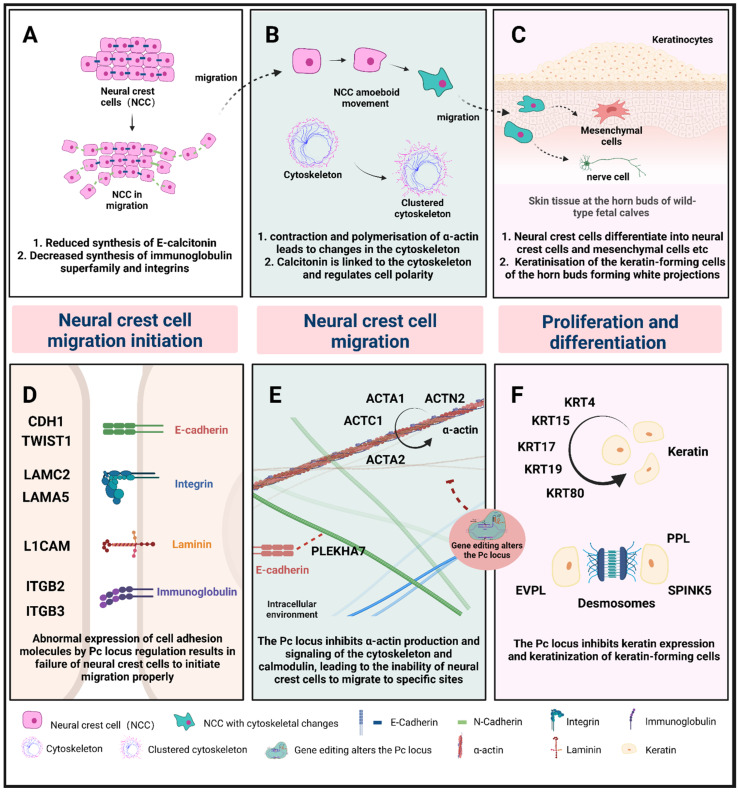
The possible mechanism by which the Pc locus regulate the polled phenotype. (**A**) Reduced expression of cell adhesion molecules such as E-calcitonin. Cells start to disperse. (**B**) The neural crest cytoskeleton changes and migrates to specific sites. (**C**) Neural crest cells differentiate into nerve cells and interstitial cells, etc., which are involved in horn formation. (**D**) Abnormal expression of cell adhesion molecules by Pc locus regulation results in the failure of neural crest cells to initiate migration properly. (**E**) The Pc locus inhibits a-actin production and signaling of the cytoskeleton and calmodulin, leading to the inability of neural crest cells to migrate to specific sites. (**F**) The Pc locus inhibits keratin expression and keratinization of keratin-forming cells.

## Data Availability

The data that support the findings of this study are available from the corresponding author, Shanjiang Zhao, upon reasonable request.

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
