# Peer review of "Uncovering Novel Features of the Pc Locus in Horn Development from Gene-Edited Holstein Cattle by RNA-Sequencing Analysis"

_ijms, 2022, doi:10.3390/ijms232012060_

Round 1

Reviewer 1 Report

Dear authors, please pay attention to the detail as follows:

Line 130 to 132, be consistent in writing the number. See 56,395,082 vs 56135334. Please fix it and check all the similar problems in the manuscript.

Line 154, 164, and 168....Fig. 8A, 8B, and 8C, should be Fig 6A, 6B, and 6C

Line 174, add a line of explanation; are the DE genes analyzed in this part the up-regulated or the down-regulated group? Or both?

Line 175 and 176. Fig 9A and 9B, change to 7A and 7B

Line 178 ...CHD1and... spacing problem, fix it and check all the similar problems in the manuscript.

Line 187. ....Fig. 10. It should be Fig 8. Seem there was a figure numbering problem in the manuscript. Please check and fix it.

Line 187. Before the expression for each gene is shown. The stable expression of the housekeeping gene in WH and EH needs to be shown. Those two groups must not be significantly different. This can be narrated in the paragraph of qRT-PCR validation or add a figure in the supplementary.

Line 196. ...a total of nine... nine or 9, please consistent

Line 214. ...compared to WH. Please add this result and refer to which figure or table. Make it in a bracket.

Line 237. The same comment is written for Line 214.

Line 246. The same comment is written for Line 214.

Line 281. The same comment is written for Line 214.

Line 299. The same comment as written for Line 214.

Line 316. Figure 9. Please elaborate on this figure in the discussion part as the hypothetical model of Pc regulation.

Line 320-321. Please describe in brief the method from the previous work [10] cited, since the reference is Chinese. Describe of i.e. method, cell line, sequence, etc, which is necessary. If not possible in the main manuscript, add all this information in the supplementary as the protocol description.

Line 350…CO2 should be CO2

Line 377...reference Roper et al., 2018, needs to be added to the reference list and follow the citation rule of IJMS. Fix this and rearrange all the reference sequences in the manuscript.

Line 384 ...shown in Table S2 was... This should be Table S1, and please complete the table with the accession number for each gene, the annealing temperature, and the PCR product size.

Line 394. ...and EH Holstein male fetal bovines.... why only the male fetus, how do you ensure that the six reconstructed embryos transferred (Line 78), resulting male fetal?

Line 430. R 4.1.2 software...add reference of this software

Line 455. Melt curve analysis in triplicate. Is this triplicate coming from three biological replicates or the technical replications? Please clear this issue.

Line 457. About the use of GAPDH as a normalizer, in table S2 GAPDH primer is not available. Moreover, table S2 needs to add one colloum of accession number for each gene. The primer sequence must be differentiated between, which is Forward (F) and reverse (R).

Line 465 to 477. Shift this and elaborate in the discussion part. See comment for line 316 above.

Author Response

Dear Editor and Reviewers,

Thank you very much for your critical review and comments on our work. This is the list of corrections of our manuscript entitled " Combinatory RNA-sequencing analysis and Tild-CRISPR/Cas9 reveals novel features of Pc locus in the Holstein horn development" submitted to IJMS. We have deal with all the comments of the reviewers. All the corrections have been marked up using the "Track Changes" function and high light in the manuscript. The whole manuscript has been carefully checked and improved. Please feel free to inform me if there are still some questions or comments.

With best regards,

Sincerely yours,

Shanjiang Zhao

Email: zhaoshanjiang@caas.cn

Response to Reviewer 1 Comments

Thank you very much for your careful review on our manuscript. Your constructive suggestions gave us a plenty of valuable information on how to write a high-quality paper. The answers to the questions are listed one by one as follows:

Point 1: Line 130 to 132, be consistent in writing the number. See 56,395,082 vs 56135334. Please fix it and check all the similar problems in the manuscript.

Response 1: We fully agree with your suggestion. According to your suggestion, we have amended "56135334" to "56,135,334" and highlight it in the manuscript (Line 135). Furthermore, we have checked all the similar problems in the manuscript. Thank you for your suggestion.

Point 2: Line 154, 164, and 168....Fig. 8A, 8B, and 8C, should be Fig 6A, 6B, and 6C

Response 2: Thank you for your advice. According to your suggestion, we have corrected the wrong picture numbers, and checked all the similar problems in the manuscript. (Lines 157, 167, and 171)

Point 3: Line 174, add a line of explanation; are the DE genes analyzed in this part the up-regulated or the down-regulated group? Or both?

Response 3: It is a good suggestion for us. The enrichment analysis in the experiment were performed based on all significantly DE genes. According to your comment, we have added a note on the up- and down-regulation of DE genes in enrichment analysis (Line 174). Thank you again for your suggestion.

Point 4: Line 175 and 176. Fig 9A and 9B, change to 7A and 7B

Response 4: Thank you for your suggestion. We have corrected the wrong picture numbers according to your suggestion, and further checked all the similar problems in the manuscript. (Lines 179, 180)

Point 5: Line 178 ...CHD1and... spacing problem, fix it and check all the similar problems in the manuscript.

Response 5: Thank you for your suggestion. We have corrected the spacing problem and checked all the similar problems in the manuscript. (Line 182)

Point 6: Line 187. ....Fig. 10. It should be Fig 8. Seem there was a figure numbering problem in the manuscript. Please check and fix it.

Response 6: We fully agree with your suggestion. According to your suggestion, we have corrected the wrong picture numbers, and further carefully checked all the similar problems in the manuscript. (Line 193)

Point 7: Line 187. Before the expression for each gene is shown. The stable expression of the housekeeping gene in WH and EH needs to be shown. Those two groups must not be significantly different. This can be narrated in the paragraph of qRT-PCR validation or add a figure in the supplementary.

Response 7: Thank you for your suggestion. We have checked the results of GAPDH for both groups and there were no significant differences. Then we performed the qRT-PCR by using GAPDH as a reference gene. Moreover, we have identified GAPDH in the manuscript. (Line 488)

Point 8: Line 196. ...a total of nine... nine or 9, please consistent

Response 8: Thank you for your suggestion. We have standardized the format of all numbers in the manuscript (Lines 188, 201). Furthermore, we have carefully checked all the similar problems in the manuscript.

Point 9: Line 214. ...compared to WH. Please add this result and refer to which figure or table. Make it in a bracket.

Response 9: It is a good suggestion for us. According to your comments, we added a note in manuscript about the source of the results for this description (Line 219).

Point 10: Line 237. Line 246. Line 281. Line 299. The same comment is written for Line 214.

Response 10: This suggestion is of great value to us. According to your comments, we have added a note in manuscript about the source of the results for this description (Lines 244, 253, 277, 302, 305).

Point 11: Line 316. Figure 9. Please elaborate on this figure in the discussion part as the hypothetical model of Pc regulation.

Response 11: Thank you for your suggestion. We have carefully thought about your comments. According to your suggestion, we have modified the name of the image and added a description of its content to the discussion (Lines 329, 336).

Point 12: Line 320-321. Please describe in brief the method from the previous work [10] cited, since the reference is Chinese. Describe of i.e. method, cell line, sequence, etc, which is necessary. If not possible in the main manuscript, add all this information in the supplementary as the protocol description.

Response 12: According to your comments, we have described the method from the previous work in brief, such as the establishment of Holstein ear margin fibroblast cell lines, experiment design and identification of sgRNAs, construction of homologous recombination, etc. (Lines 339-352)

Point 13: Line 350…CO2 should be CO2

Response 13:Thank you for your suggestions. We have corrected the error in writing " CO2" and highlight it (Line 381). Furthermore, we have carefully proofread the manuscript.

Point 14: Line 377...reference Roper et al., 2018, needs to be added to the reference list and follow the citation rule of IJMS. Fix this and rearrange all the reference sequences in the manuscript.

Response 14: Thank you for your suggestion. According to your advice, we have corrected "reference Roper et al., 2018" to the correct format according to the format of the reference (Line 407).

Point 15: Line 384 ...shown in Table S2 was... This should be Table S1, and please complete the table with the accession number for each gene, the annealing temperature, and the PCR product size.

Response 15: According to you suggestion, we have added the accession number for each gene, the annealing temperature, and the PCR product size. Moreover, in order to show the DEGs, we added a new table as Table S1. Thank you again for your suggestion.

Point 16: Line 394. ...and EH Holstein male fetal bovines.... why only the male fetus, how do you ensure that the six reconstructed embryos transferred (Line 78), resulting male fetal?

Response 16: Thank you very much for your invaluable suggestion on our manuscript. In our experiments, the nuclear donor cells were derived from the Holstein bull ear margin fibroblast cell lines, so after somatic cell cloning all embryos were male. Furthermore, the phenotypic results indicated that the genital phenotype of the fetal bovine was male too. Therefore, we selected wild-type Holstein male fetal bovine as the control group.

Point 17: Line 430. R 4.1.2 software...add reference of this software

Response 17: Thank you for your suggestion. We have added the appropriate references in the manuscript. (Lines 461, 465)

Point 18: Line 455. Melt curve analysis in triplicate. Is this triplicate coming from three biological replicates or the technical replications? Please clear this issue.

Response 18: Thank you for your suggestion. In this experiment, the melt curve analysis for the WH group came from three biological replicates. In EH group, one pregnancy was terminated on day 90 of gestation for analysis of the fetus. After confirming that the fetal bovine was a polled phenotype by genomic testing and phenotypic identification, we performed technical replicates.

Point 19: Line 457. About the use of GAPDH as a normalizer, in table S2 GAPDH primer is not available. Moreover, table S2 needs to add one colloum of accession number for each gene. The primer sequence must be differentiated between, which is Forward (F) and reverse (R).

Response 19: Thank you for your suggestion. We have added the primers for GAPDH in table S2 and add one colloum of accession number for each gene according to your suggestion. In addition, we added markers for forwarding and reverse primers to the table. Thank you again for your suggestion. (Now Table S2)

Point 20: Line 465 to 477. Shift this and elaborate in the discussion part. See comment for line 316 above.

Response 20: It is a good suggestion for us. According to your comments, we have adjusted this section to the discussion section for elaboration (Lines 320-334). Thank you again for your suggestion.

Reviewer 2 Report

Overview

Using SCNT authors obtained genome edited calf fetuses with homozygous Polled genotype and performed transcriptome analysis on horn bud tissues of edited and wt (horned) fetuses. Authors identified several differentially expressed genes, mostly related to cell adhesion, actin cytoskeleton and keratinization pathways.

Major issues

Authors provide data on analysis of 3-month-old fetuses obtained by slaughter. Why authors did not get any newborn calves after blastocyst transfer? Fetuses seemed to develop normally (fig 2), so it is not clear why authors slaughtered pregnant cows and performed experiments on fetuses instead of waiting for the newborn calves. Or alternatively, more blastocysts could be transferred so authors could analyze both fetuses and newborn calves.

Transcriptome analysis was performed only at one genome-edited fetus, so these results should be interpreted with extreme caution. Sample size should be at least 3 to draw a statistically significant conclusion.

Authors provide detailed discussion of cell adhesion and EMT mechanisms which presumably underlie horn development. In addition, authors speculate the Polled genotype might inhibit EMT of neural crest cell thereby inhibiting horn formation. This hypothesis was not supported experimentally. In my opinion, such detailed discussion of these mechanisms lies out of the scope of the manuscript.

Specific points

WH and EH abbreviations could be changed to Horned/Polled for clarity. Fig.1.

Fig1: Panels A and B do not provide any relevant data and should be removed.

Fig1D – Actual Sanger sequencing chromatograms should be presented.

Fig4 is just the repeat of Fig1C and D and should be removed.

Panels in Fig3 should be labeled Horned/Polled for clarity.

There is a mistake in Fig3 legend. Is says about 2-3 layers of cells for both Horned/Polled phenotype. Text of the manuscript refers to 10 layers of cells for Horned phenotype.

Fig8 legend is a non-informative.

Text in section 2.4 refers to Fig8C but the data are shown on Fig6.

Text in section 2.6 refers to Fig10, but there is no Fig10 in the manuscript.

In the discussion section authors mistakenly named N-cadherin and E-cadherin which are key players in EMT, as “N-calmodulin” and “E-calmodulin” (see page 10, line 226).

Author Response

Dear Editor and Reviewers,

Thank you very much for your critical review and comments on our work. This is the list of corrections of our manuscript entitled " Combinatory RNA-sequencing analysis and Tild-CRISPR/Cas9 reveals novel features of Pc locus in the Holstein horn development" submitted to IJMS. We have deal with all the comments of the reviewers. All the corrections have been marked up using the "Track Changes" function and high light in the manuscript. The whole manuscript has been carefully checked and improved. Please feel free to inform me if there are still some questions or comments.

With best regards,

Sincerely yours,

Shanjiang Zhao

Email: zhaoshanjiang@caas.cn

Response to Reviewer 2 Comments

Thank you very much for your kind words about our work and for your valuable suggestions on the manuscript, which are very important to us. We have carefully revised the article according to each of your suggestions. The answers to the questions are listed one by one as follows:

Point 1: Authors provide data on analysis of 3-month-old fetuses obtained by slaughter. Why authors did not get any newborn calves after blastocyst transfer? Fetuses seemed to develop normally (fig 2), so it is not clear why authors slaughtered pregnant cows and performed experiments on fetuses instead of waiting for the newborn calves. Or alternatively, more blastocysts could be transferred so authors could analyze both fetuses and newborn calves.

Response 1: This is a good suggestion for our future research. In 2013, Allais et al. described the wild type horn buds can be detected at the gestation day 90(GD90). In 2015, Wiener et al. also reported that the potential horn buds can be detected at an early stage of development. In the horned wild-type control, histological analysis of the horn buds revealed thickening of the epidermis with 11-13 layers of vacuolated keratinocytes. Moreover, no fetal hair follicles were detected in the dermal layers beneath the horn bud. In contrast, in the fetus carrying the Pc variant, no horn buds were detected macroscopically. The histological examination showed only slight epidermal thickening with two to three layers of vacuolated keratinocytes. Recently, Schuster et al (2020) also chose the point of GD90 to conduct their study, one pregnancy was terminated on day 90 of gestation for analysis of the fetus. Taken together, in our study, we slaughtered fetal cows obtained at 90 days after embryo transfer to explore the possible molecular mechanisms by which Pc regulates the polled. And the rest of the fetal cows are continuing to develop, the gestation period is currently at around 180 days. We will continue embryo transfer later when the COVID-19 epidemic has subsided. Thank you again for your suggestion.

Point 2: Transcriptome analysis was performed only at one genome-edited fetus, so these results should be interpreted with extreme caution. Sample size should be at least 3 to draw a statistically significant conclusion.

Response 2: This suggestion is of great value to us. Before transcriptome analysis, in order to clarify the polled phenotype, we performed PCR identification, genome sequencing, and HE staining of 90-day fetal bovine horn buds. On the basis of the clear polled phenotype and consideration of the higher rate of large-age abortions in gene-edited cattle, one pregnancy was terminated on day 90 of gestation for analysis of the fetus considering. In future, we will continue to increase the number of embryo transferred, wants to acquire more experimental materials.

Point 3: Authors provide detailed discussion of cell adhesion and EMT mechanisms which presumably underlie horn development. In addition, authors speculate the Polled genotype might inhibit EMT of neural crest cell thereby inhibiting horn formation. This hypothesis was not supported experimentally. In my opinion, such detailed discussion of these mechanisms lies out of the scope of the manuscript.

Response 3: Thank you for your suggestion. We have carefully thought about your comments. Considering the specificity and difficulty of horn studies, we have reviewed the literature related to horn development as comprehensively as possible and combined the results of the DEGs enrichment and protein interaction analysis, we have tried to make speculative hypothesis on the developmental mechanism of horn especially in the embryonic stage. We hope that it will provide a reference or future research direction for Pc regulation of the polled trait in Holstein. Thank you again for your valuable comments, and we will continue to conduct in-depth research on the proposed hypothesis in the future.

Point 4: WH and EH abbreviations could be changed to Horned/Polled for clarity. Fig.2.

Response 4: This suggestion is of great value to us. Based on your suggestion, we have revised the naming of the groups in Fig.2 (Line 112). Thank you again for your advice.

Point 5: Fig1: Panels A and B do not provide any relevant data and should be removed.

Response 5: Thank you for your suggestion. Panels A and B are our actual photographs. Considering that Panel B can reflect the quality of embryo development to a certain extent, it is perhaps better to keep it for the time being.

Point 6: Fig1D – Actual Sanger sequencing chromatograms should be presented.

Response 6: It is a good suggestion for us. According to your comment, we modified Fig1D and replaced it with Actual Sanger sequencing chromatograms (Line 85).

Point 7: Fig4 is just the repeat of Fig1C and D and should be removed.

Response 7: Thank you for your suggestion. We have carefully thought about your comments. Fig 1C and D showed the results of the embryo genome testing, while Fig 4 showed the results of the genomic testing of the gene-edited fetal bovine, so it is perhaps better to keep it for the time being.

Point 8: Panels in Fig3 should be labeled Horned/Polled for clarity.

Response 8: Based on your suggestion, we have labeled Horned/Polled in the figure legend on the left in Fig3 (Line 114). Thank you for your advice.

Point 9: There is a mistake in Fig3 legend. Is says about 2-3 layers of cells for both Horned/Polled phenotype. Text of the manuscript refers to 10 layers of cells for Horned phenotype.

Response 9: Thank you very much for your careful review on our manuscript. It is possible that the unclear description in the diagram has caused you to misunderstand. Formerly, we only detailed descriptions of the horned frontal skin and the polled horn bud skin in the figure legend. Both of these groups only have 2-3 layers of cells. With regard to the problem you mention, we have added a detailed description of the horned horn buds in the figure legend to make it much clearer (see figure below). (Line

117)

Figure 3. Histological identification of the fetus. (A) (B) (C) (D) is wild-type frontal skin tissue with normal thickness epidermis, only 2-3 layers of vacuolated keratin-forming cells, epidermis with follicular growth, no thickened nerve bundles found under the skin. (E) (F) (G) (H) is wild-type skin tissue at the horn bud with more than 10 layers of vacuolated keratin-forming cells and no follicular growth in the epidermis. There are thickened nerve bundles under the skin. (I) (J) (K) (L) is skin tissue at the horn buds of polled cattle, with a normal thickness of epidermis visible, only 2-3 layers of vacuolated keratin-forming cells, and no thickened nerve bundles found under the horn buds. (A) (I) magnification X3, (E) magnification X5, (B) (C) (F) (G) (J) (K) magnification X10, (D) (H) (L) magnification X30.

Point 9: Fig8 legend is a non-informative.

Response 9: It is a good suggestion for us. We have described fig8 in detail (The red line in the graph shows the results of RT-qPCR, the blue line is the result of RNA-seq) and highlight it in the manuscript. (Line 193)

Point 10: Text in section 2.4 refers to Fig8C but the data are shown on Fig6.

Response 10: Thank you for your suggestion. We have corrected the error and apologize for it (Lines 167,171). Thank you for your advice!

Point 11: Text in section 2.6 refers to Fig10, but there is no Fig10 in the manuscript.

Response 11: Thank you for your suggestion. We have corrected the wrong picture numbers. Furthermore, we have checked all the similar problems in the manuscript. Thank you for your suggestion. (Line 191)

Point 12: In the discussion section authors mistakenly named N-cadherin and E-cadherin which are key players in EMT, as “N-calmodulin” and “E-calmodulin” (see page 10, line 226).

Response 12: Thank you very much for your careful review on our manuscript. This was a low-level mistake. We corrected the misrepresentation and carefully proofread the manuscript. (Lines 231, 246, 247, 260, 283)

Round 2

Reviewer 2 Report

Authors have carefully replied to all previous comments, mostly correcting the errors in the preparation of the manuscript. Nevertheless, I still have several concerns about the submitted manuscript.

Comments

1.      Did authors check the established cell line for off-target integration of DNA repair template? Authors used Tild-CRISPR and the linear DNA fragments are prone to random integration in the genome. The original article by Wang et al (ref 10) is not accessible.

2.      The title of the article includes “Tild-CRISPR/Cas9”. Actually, herein authors did not use this approach, the genome-edited cell line was obtained in the previous work, that was cited in this manuscript. I think that the title should be changed and the “Tild-CRISPR/Cas9” term should be removed for clarity.

3.      Fig1D with real Sanger chromatograms is of a poor quality.

4.      Fig4B – the same notion as above, Sanger chromatograms should be provided. For clarity, the text representation of nucleotide sequencing results could be used as well.

5.      Authors discuss the molecular mechanisms of horn development and it is clear that changes in gene expression are involved in this process in the Polled genotype animals. Unfortunately, it is not clear how the Polled locus can affect all the discussed mechanisms of horn development at the molecular level. What is encoded in the Polled locus? Are there any regulatory sequences, genes, long non coding RNAs, enhancers etc? What genes could be affected by the Polled duplication in the first place and what is the presumable mechanism? Another possibility, is the perturbation of long-distance interactions in the genome by the Polled duplication etc. All these mechanisms are important to discuss.

Author Response

Response to Reviewer 2 Comments

Thank you very much for your invaluable suggestions on our manuscript, which are very important to us. We have carefully revised the article according to each of your suggestions. The answers to the questions are listed one by one as follows:

Point 1: Did authors check the established cell line for off-target integration of DNA repair template? Authors used Tild-CRISPR and the linear DNA fragments are prone to random integration in the genome. The original article by Wang et al (ref 10) is not accessible.

Response 1: Thank you very much for your valuable suggestions on the manuscript, which are very important to us. Firstly, we checked the article published by Yao et al. (2018) [1]. The article stated that the targeting efficiency of knockin sequences via homologous recombination (HR) is generally low. Therefore, they describe a method they call Tild-CRISPR (targeted integration with linearized dsDNA-CRISPR), a targeting strategy in which a PCR-amplified or precisely enzyme-cut transgene donor with 800-bp homology arms is injected with Cas9 mRNA and single guide RNA into mouse zygotes. Compared with existing targeting strategies, this method achieved much higher knockin efficiency in mouse embryos, as well as brain tissue. Importantly, the Tild-CRISPR method also yielded up to 12-fold higher knockin efficiency than HR-based methods in human embryos, making it suitable for studying gene functions in vivo and developing potential gene therapies. Notably, considering that linearized double-strand DNA (dsDNA) as templates in Tild-CRISPR may result in more random insertions than using equal doses of circular DNAs in homologous recombination (HR) - or homology-mediated end-joining (HMEJ)-based methods, Yao et al. (2018) further performed Southern blot analysis of the samples from the mice bearing mCherry knockin at Cdx2 locus by the HMEJ-based method and Tild-CRISPR. By using mCherry internal probe, additional randomly integrated transgenes were detected in 2 out of 7 mCherry knockin fetuses by the HMEJ-based method and in 2 out of 8 mCherry knockin mice by Tild-CRISPR. Compared with HMEJ-based and HR-based methods, the random insertion rate of Tild-CRISPR was not significantly increased [1]. Together, these results indicate that the Tild-CRISPR method yielded robust DNA integration, and it may be a usable assay which not only could increase editing accuracy, but also can maintain lower off-target effects (See Fig1 below). Secondly, in our experiment, the cell lines we prepared with homozygous Pc inserts grew normally, and the polled fetal bovine also showed normal development all round, with no abnormal phenotypes seen that might have been brought by the off-target. Thirdly, in a previous experiment, we used Cas-OFFinder software to perform predictive analysis of off-target sites for sgRNAs, and the results showed that No high-risk off-target sites were found (See Fig 2 below). Thank you again for your suggestion. Fig. 1 Southern blot analysis of Cdx2-p2A-mCherry mice generated by Tild-CRIPSR. (A) Schematic overview of experimental design. Internal mCherry probe were indicated by red bars. Green bar, homology arm; F/R, forward/reverse primer. (B) Genotyping results of Cdx2-p2A-mCherry fetus or mice used for southern blot. Seven knock-in fetus generated by HMEJ-based method and eight knock-in mice generated by Tild-CRIPSR were randomly selected. (C) EcoRI-digested genomic DNA from Cdx2-p2A-mCherry mice were hybridized with internal mCherry probe. Internal mCherry probe expected fragment size: WT = N/A, Targeted = 4.2 kb. White asterisk, random integration. M, marker. Fig.2 The result of Cas-OFFinder assay of the Pc sgRNA

Related reference
[1] Yao X, Zhang M, Wang X, Ying W, Hu X, Dai P, Meng F, Shi L, Sun Y, Yao N, Zhong W, Li Y, Wu K, Li W, Chen ZJ, Yang H. Tild-CRISPR Allows for Efficient and Precise Gene Knockin in Mouse and Human Cells. Dev Cell. 2018 May 21; 45(4):526-536.e5. doi: 10.1016/j.devcel.2018.04.021. PMID: 29787711.

Point 2: The title of the article includes “Tild-CRISPR/Cas9”. Actually, herein authors did not use this approach, the genome-edited cell line was obtained in the previous work, that was cited in this manuscript. I think that the title should be changed and the “Tild-CRISPR/Cas9” term should be removed for clarity.

Response 2: We fully agree with your suggestion. According to your comment, we have changed the title of the article to “Uncovering novel features of Pc locus in horn development from the gene-edited Holstein cattle by RNA-sequencing analysis” (Lines 1-3). Thanks for your advice.

Point 3: Fig1D with real Sanger chromatograms is of a poor quality.

Response 3: It is a good suggestion for us. According to your comment, we have improved the quality of fig 1D. Moreover, we have also adjusted the distribution of the results in Fig1 to make the picture clearer (Lines 91, 97). Thank you for your suggestion.

Point 4: Fig4B – the same notion as above, Sanger chromatograms should be provided. For clarity, the text representation of nucleotide sequencing results could be used as well.

Response 4: Thank you for your suggestion. According to your advice, we have adjusted the fig4B to a sanger chromatogram (Lines 138, 142, 143). Thank you again for your suggestion.

Point 5: Authors discuss the molecular mechanisms of horn development and it is clear that changes in gene expression are involved in this process in the Polled genotype animals. Unfortunately, it is not clear how the Polled locus can affect all the discussed mechanisms of horn development at the molecular level. What is encoded in the Polled locus? Are there any regulatory sequences, genes, long non coding RNAs, enhancers etc? What genes could be affected by the Polled duplication in the first place and what is the presumable mechanism? Another possibility, is the perturbation of long-distance interactions in the genome by the Polled duplication etc. All these mechanisms are important to discuss.

Response 5: So far, we have carefully searched all the literature related to the Pc locus. Possibly due to the specificity of horn development and the difficulty of research, since Pc was first reported, most studies of the Pc locus were still focused on phenotypic research (Association of the presence or absence of the Pc locus with horned/polled traits). In 2013, when studying horn buds, Allais-Bonnet et al. observed a tissue-specific overexpression of LincRNA#1 in Pc/p horn buds with significant differences (p,0.05) vs. Pc/p or wt frontal skin, and suggestive differences (p = 0.052) vs. wt horn buds [1]. In 2022, Hennig et al. investigated the regulatory relationship between lincRNA#1 and the Pc locus. Unfortunately, the results of the study showed thatlincRNA#1 knockout fetuses revealed similar morphology to non-edited, control polled fetuses, indicating the absence of lincRNA#1 alone does not result in a horned phenotype [2]. Thus, in order to reveal the underlying biological mechanism causing polled. After performing PCR identification, genome sequencing, and HE staining to clarify the hornless phenotype of the fetal cow, we performed the RNA-seq on the horn buds of polled fetal bovine for the first time. At the same time, we referred to the results of Wang Yu et al. on the origin of horn development [3]. Based on the above reasons, we developed the hypothesis. We hope to show the possible molecular mechanisms by which the Pc locus regulates the polled trait. Furthermore, we have studied your comments carefully. Your ideas on the regulation of the polled trait by the Pc locus are very enlightening to us, for example, what is encoded in the Polled locus? Are there any regulatory sequences, genes, long non coding RNAs, enhancers, etc. In the future, we will continue to investigate the molecular mechanism of the Pc locus regulating the hornless trait based on your suggestions. Thank you again for your valuable advice, which has been of great help to us.

Related references
[1] Allais-Bonnet A, Grohs C, Medugorac I, Krebs S, Djari A, Graf A, Fritz S, Seichter D, Baur A, Russ I, Bouet S, Rothammer S, Wahlberg P, Esquerré D, Hoze C, Boussaha M, Weiss B, Thépot D, Fouilloux MN, Rossignol MN, van Marle-Köster E, Hreiðarsdóttir GE, Barbey S, Dozias D, Cobo E, Reversé P, Catros O, Marchand JL, Soulas P, Roy P, Marquant-Leguienne B, Le Bourhis D, Clément L, Salas-Cortes L, Venot E, Pannetier M, Phocas F, Klopp C, Rocha D, Fouchet M, Journaux L, Bernard-Capel C, Ponsart C, Eggen A, Blum H, Gallard Y, Boichard D, Pailhoux E, Capitan A. Novel insights into the bovine polled phenotype and horn ontogenesis in Bovidae. PLoS One. 2013 May 22; 8(5):e63512. doi: 10.1371/journal.pone.0063512.
[2] Hennig SL, McNabb BR, Trott JF, Van Eenennaam AL, Murray JD. LincRNA#1 knockout alone does not affect polled phenotype in cattle heterozygous for the celtic POLLED allele. Sci Rep. 2022 May 10; 12(1):7627. doi: 10.1038/s41598-022-11669-9. PMID: 35538091; PMCID: PMC9090918.
[3] Wang Y, Zhang C, Wang N, Li Z, Heller R, Liu R, Zhao Y, Han J, Pan X, Zheng Z, Dai X, Chen C, Dou M, Peng S, Chen X, Liu J, Li M, Wang K, Liu C, Lin Z, Chen L, Hao F, Zhu W, Song C, Zhao C, Zheng C, Wang J, Hu S, Li C, Yang H, Jiang L, Li G, Liu M, Sonstegard TS, Zhang G, Jiang Y, Wang W, Qiu Q. Genetic basis of ruminant headgear and rapid antler regeneration. Science. 2019 Jun 21; 364(6446): eaav6335. doi: 10.1126/science. aav6335. PMID: 31221830.
